# A Critical Study of Using the Peukert Equation and Its Generalizations for Determining the Remaining Capacity of Lithium-Ion Batteries

**Nikolay E. Galushkin \*** , **Nataliya N. Yazvinskaya and Dmitriy N. Galushkin**

Laboratory of Electrochemical and Hydrogen Energy, Don State Technical University, 346500 Town of Shakhty, Rostov Region, Russia; lionnat@mail.ru (N.N.Y.); dmitrigall@yandex.ru (D.N.G.)
**\*** Correspondence: galushkinne@mail.ru; Tel.: +7-9-2876-978-20



**Featured Application: lithium-ion batteries.**

**Abstract:** In many papers for forecasting remaining capacity of lithium-ion batteries, various analytical models are used based on the Peukert equation. In this paper, it is shown that the classic Peukert equation is applicable in two ranges of discharge currents. The first range isis the battery released capacity and ) to currents at which the discharge capacity of battery begins to rapidly decrease. The second range of discharge currents is from the inflection point of experimental curve to the highest currents used in the experiments. In the first range of discharge currents, both the classic Peukert equation and the Liebenow equation can be used. The operating range of the discharge currents for commercial automotive-grade lithium batteries is in the first range. Therefore, in many of the analytical models, the classic Peukert equation (taking into account the temperature) is successfully used to estimate the remaining capacity of these batteries. An analysis and evaluation of advantages and disadvantages of all the most popular generalized Peukert equations is presented. The generalized Peukert equation with allowance for temperature is established, which makes it possible to estimate the released capacity with high accuracy for lithium-ion batteries.

**Keywords:** Peukert equation; cell; lithium-ion; capacity; discharge current; state-of-charge

---

## 1. Introduction

From the creation of lithium-ion batteries until now, their scope of use has constantly expanded, and they often supersede the batteries of other electrochemical systems (both alkaline and acidic). First, this is connected with their high specific capacity and power. At present, lithium-ion batteries dominate in the segment of small-format batteries (smartphones, household appliances, etc.) [1,2].

In recent years, in connection with the severe environmental situation (first, in large cities), the intensive development of environmentally clean transportation means started: hybrid electric vehicles (HEVs), battery electric vehicles (BEVs) and plug-in hybrid electric vehicles (PHEVs) [2–5].

However, for adequate battery management in different technical systems, a reliable battery model is necessary. In addition, the model must be able to provide an accurate evaluation of the state of charge (SoC), as this parameter largely determines the performance of the whole system.

Many methods and models for the SoC evaluation have been developed and proposed. The simplest method of the SoC evaluation is based on the open circuit voltage [6]. However, this method is not accurate because of the relaxation processes, especially under a dynamic load (error of 20% or more) [7]. In addition, this method is not suitable for batteries with a flat discharge curve, in particular for lithium-iron-phosphate batteries (LFP) [8]. Another widespread method of SoC

evaluation is using models based on the Kalman filter and fuzzy logic [8–13]. These models provide much better SoC evaluation [11] than the previous methods, especially at dynamic loads. However, in our experiments, the relative error of SoC evaluation by this method was about 10%, which is quite high.

The modern method of SoC evaluation relies on the combination of the following two methods: counting of ampere hours and use of voltage profiles. However, this method makes the SoC calculation a difficult problem, and it has known accuracy problems [1].

From the theoretical point of view, the most accurate battery models can be obtained only with use of the electrochemical laws of molecules and ions transport with due account of all the internal characteristics of batteries, i.e., with use of the electrochemical method of modeling [14–17]. However, for their service, these models require powerful computing systems not typical for modern mobile vehicles and appliances. Thus, they are unacceptable for electric vehicles, airplanes and other mobile systems containing the batteries. They are inconvenient to use in practice, too, as they require a long calibration and finding out of all internal parameters of a battery [18].

An alternative to the above-described methods can be the analytical models based on various generalizations of the Peukert equation [18–21] or on other empiric equations [22]. However, even when modeling batteries by the electrochemical method, one is often forced (within the frameworks of those models) to use additional analytical models when it is necessary to take into account such poorly studied processes as the thermal runaway [23–25], the accumulation of hydrogen in battery electrodes [26,27] and the gas generation during lithium-ion battery cycling [28,29].

Analytical models are often used for SoC evaluation in the lithium-ion batteries [18,19]; moreover, the temperature of the batteries is also taken into account. We believe the most promising method is the analytical model of Hausmann [18]. This model is specifically designed to evaluate the remaining capacity of batteries in electric vehicles. This model is based on the classic Peukert equation. However, the classic Peukert equation does not correspond to experimental data for lithium-ion batteries at any discharge current.

The purpose of this paper is to find the ranges of discharge currents in which the classic Peukert (and Liebenow) equation is applicable for lithium-ion batteries, since the classic Peukert equation is very often used in various analytical models of lithium-ion batteries [18]. In addition, this paper aims at weighing the advantages and disadvantages of the most popular generalized Peukert equations and choosing an optimal equation for SoC evaluation in the lithium-ion batteries (at any discharge current), for example, based on the analytical models proposed in [18,19].

## 2. Generalized Peukert's Equations

Consider the most well-known empirical equations for calculation of the batteries released capacity at different discharge currents:

- Peukert equation [30]

$$C = \frac{A}{i^n} \quad (1)$$

- Liebenow equation [31]

$$C = \frac{A}{1 + n \cdot i} \quad (2)$$

- generalized Peukert equations [32,33]

$$C = \frac{A}{i^n} \tanh\left(\left(\frac{i}{B}\right)^n\right) \quad (3)$$

$$C = \frac{A}{1 + B \cdot i^n} \quad (4)$$

where $i$ is the discharge current, $C$ is the battery released capacity and $A$, $B$ and $n$ are the empiric constants. It should be noted that Equations (1) and (2) were obtained based on the experimental

research of lead-acid batteries. However, afterwards, they were used for capacity evaluation of other types of batteries, too (e.g., lithium-ion [18,34]). Of course, there exist other calculation methods for the battery released capacity [35–37]; however, as a rule, they are either a combination or particular case of Equations (1)–(4).

The battery discharge process is the phase transition; however, the phase transitions are often described by the complementary error function [38]

$$C(i) = \frac{A}{2} \cdot erfc\left(\frac{i - i_0}{\sqrt{2}\sigma}\right) \tag{5}$$

where $\sigma$ is the standard deviation and $i_0$ is the mean value of the statistical variable $i$. That is why we are investigating this equation too.

## 3. Experimental

In our experiments, we used lithium-ion cells of various manufacturers, capacities and formats (the manufacturers, models and nominal capacities of the studied cells are presented in Tables 1 and 2).

The charge procedure consisted of using the constant current/constant voltage (CC/CV) mode. The charge was conducted as follows: the direct current $0.5C_n$ to a voltage value of 4.2 V and step CV until the current falls to $0.025C_n$ ($C_n$ is the nominal cell capacity).

The discharge procedure consisted of applying the constant current (CC) mode. The discharge was conducted as follows: the direct current was applied down to the voltage value 2.75 V. In the course of the cells study, the discharge currents were in the range from $0.2C_n$ to the currents at which the released capacity was approximately $C \approx C_n/10$. The cells were cycled inside of the climatic chamber Binder MK240 (BINDER GmbH, Germany).

Experimental studies of the cells were performed in accordance with the following algorithm.

First, the cells were cycled at least ten times to stabilize cell parameters in connection with the SEI layer formation. The cycling was stopped when, in three consecutive cycles, the measured capacity differed by less than 5%. These training cycles were implemented in accordance with the cell's operation manual (charge: the standard mode described above; discharge: direct current equal to $0.2C_n$ to a voltage value of 2.75 V).

Second, to decrease a random scatter of the measured cell capacity, the discharge capacity of the cell (at a certain discharge current) was obtained as the average value of the measured capacities obtained for three consecutive charge–discharge cycles. Nevertheless, if in these three cycles the measured capacity differed more than by 5%, additional training cycles were implemented and the experiment was repeated again (or the cell was replaced with another more stable cell and in this case the experiment was implemented from the very beginning).

Third, to prevent a mutual influence of the charge–discharge cycles (via various residual phenomena), before each value change of the discharge current or the cell temperature, the training cycles were implemented. The training cycles were performed until the measured capacity in three consecutive cycles began to differ by less than 5%.

In any batch of the same cells, statistical scatter exists of cells measured capacities, which is attributed to many random factors such as the cell's parameter's statistical scatter in the course of the cells production, the cell's operation duration, the cell's operation modes, etc. To decrease this random scatter as much as possible, we standardized the experimental cell's obtained capacities by their top capacity $C_m$ (which was found experimentally for every cell individually). In this case, the above listed random factors were eliminated to a large extent and the empiric curves were constructed more reliably. This method is used in Section 4.2 to study the dependence of cell released capacity on discharge current.

## 4. Results and Discussion

The cell released capacity depends on the cell temperature [13,39–41]. As cell temperature lowers, the cell released capacity decreases because of slowing down of the chemical reactions [13]. This fact is ignored completely in the classic Peukert Equation (1). However, it can result in considerable errors in the case of an ambient temperature change or self-heating effects in the cell [42].

### 4.1. Dependence of Released Capacity on Cell Temperature

Consider the temperature effect to the released capacity of the lithium-ion cells. The cells were cycled in the training mode inside of the climatic chamber Binder MK240 (BINDER GmbH, Germany) at various temperatures. The cell temperature was determined using LM35 temperature sensor attached to the cell. The obtained experimental results are given in Figure 1. In the experiments, batteries were used with $LiMn_2O_4$ (IMR) and $LiNiMnCoO_2$ (INR) cathodes.

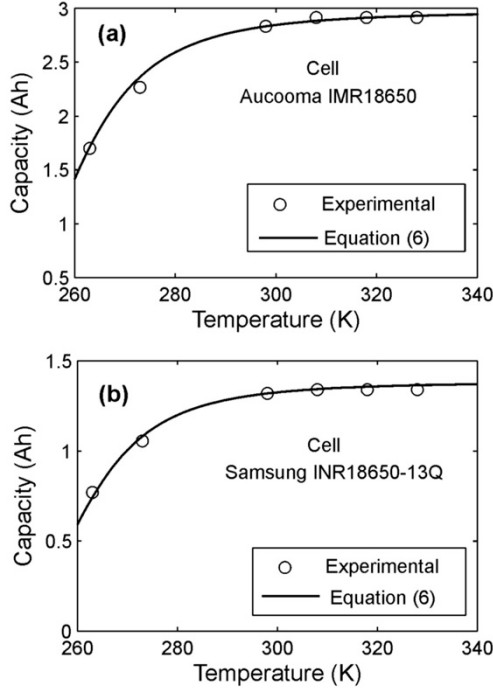

**Figure 1.** Dependence of cell released capacity on temperature of studied cells at discharge current 0.2 $C_n$ ($C_n$ is cell nominal capacity): (**a**) cell with cathode $LiMn_2O_4$ (IMR); and (**b**) cell with cathode $LiNiMnCoO_2$ (INR).

According to the authors of [19], the dependence of the released by lithium-ion cells capacity on temperature is described by the equation

$$C = C_{mref} K \frac{\left(\frac{T-T_L}{T_{ref}-T_L}\right)^{\beta}}{(K-1)+\left(\frac{T-T_L}{T_{ref}-T_L}\right)^{\beta}} \tag{6}$$

where $T_{ref}$ is a reference temperature of cell, $C_{mref}$ is top capacity released at temperature $T_{re}$ and $T_L$ is a temperature at which $C = 0$, i.e., temperature, at which all the electrochemical processes stop.

In the temperature range from $T_{ref}$ to infinity, the capacity $C$ in Equation (6) changes from $C_{mref}$ to $C_{mref}K$, thus parameter $K$ shows how many times (theoretically) the capacity $C$ can be increased with the increase in cell temperature.

Based on the obtained experimental data (Figure 1), the optimal parameters for Equation (6) were found by the least square method and the Levenberg–Marquardt optimization procedure. The optimal parameters are listed in Table 1.

According to the conducted research (Figure 1), in the temperature range 25–55 °C, the relative deviation of capacity from its average value is less than 1%. Thus, in this temperature range, it is possible to neglect the influence of temperature when studying the dependence of the cell released capacity on the discharge current. This statement is correct for all cells studied by us (Tables 2–4).

**Table 1.** Optimal values of parameters for Equation (6).

| Model | $T_{ref}$ (K) | $C_{mref}$ (Ah) | $\beta$ | $T_L$ (K) | $K$ | $\delta^1$ (%) |
|---|---|---|---|---|---|---|
| Aucooma IMR18650 | 298 | 2.8 | 3.012 | 239.022 | 1.049 | 2.001 |
| Samsung INR18650-13Q | 298 | 1.3 | 3.101 | 240.225 | 1.048 | 2.512 |

[1] Relative error of experimental data approximation by Equation (6) in Figure 1.

*4.2. Dependence of Cell Released Capacity on Discharge Current*

To decrease the temperature changes during cell discharge by high currents, we used a number of methods. Firstly, cells were cycled at the temperature T = 25 °C inside of the climatic chamber Binder MK240 (BINDER GmbH, Germany). Secondly, modified heat sinks were used (usually applied for refrigeration of processors in computing systems). The heat sinks were fastened to the cells using a specially manufactured clamp and MX-2 heat-conducting paste produced by ARTIC. This arrangement greatly increased the thermal flux from the cells and accelerated their refrigeration. These measures allowed keeping the cell temperature below 55 °C in all our experiments.

It should be noted that all the cells studied by us had no protection; therefore, it was possible to discharge them with high currents.

For convenience of comparison between the experimental data and Equations (3)–(5), we rewrite them in the following form

$$C = \frac{C_m}{(i/B)^n} \tanh\left(\left(\frac{i}{B}\right)^n\right) \tag{7}$$

$$C = \frac{C_m}{1 + \left(\frac{i}{B}\right)^n} \tag{8}$$

$$C = \frac{C_m}{\text{erfc}(-B/n)} \text{erfc}\left(\frac{i-B}{n}\right) \tag{9}$$

For Equations (7)–(9), the condition $C(0) = C_m$ is true, i.e., $C_m$ is the maximum cell capacity.

The obtained experimental data are given in Figure 2. In Figure 2, the experimental data are compared with Equation (9), as this equation had the least relative error of approximation of the experimental data in all our experiments (Table 2).

Based on the obtained experimental data (Figure 2), the optimal parameters for Equations (7)–(9) were found using the least square method and the Levenberg-Marquardt optimization procedure. The optimal parameters are shown in Table 2.

Standardization of the released capacity and the cell discharge current by the top capacity reduces the investigation of a particular cell to studying the cell with unit capacity. Hence, in the standardized coordinates, the cells should have the same experimental function $f(i) = C(i/C_m)/C_m$, provided that the electrodes and the electrolyte are the same for these cells. For example, in Figure 2a, the experimental functions $f(i) = C(i/C_m)/C_m$ for the cells TrustFire IMR16340 and Aucooma IMR18650 coincide within the limits of the experimental error. Notwithstanding that these cells differ with their capacity and format, as well as are made by different manufacturers, they have the same type of cathodes (IMR).

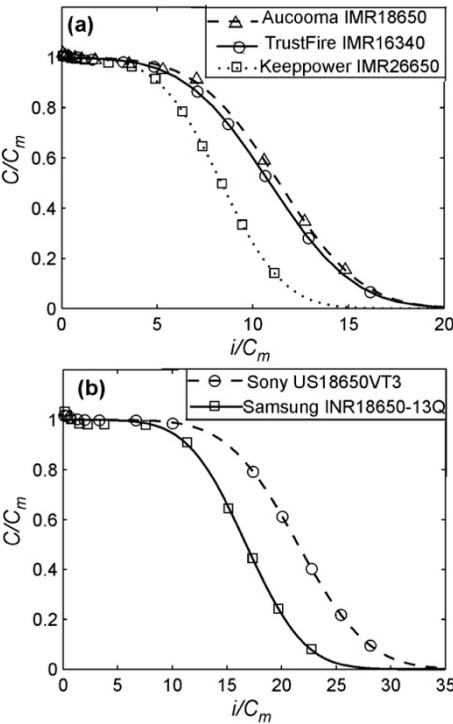

**Figure 2.** Comparison between experimental data for lithium-ion cells of different types and Equation (9): (**a**) cell with cathode $LiMn_2O_4$ (IMR); and (**b**) cell with cathode $LiNiMnCoO_2$ (INR). For each cell, the value of $C_m$ was taken from Table 2. △, ○, □, experimental values.

**Table 2.** Optimal parameters of generalized Peukert Equations (7)–(9) for lithium-ion cells.

| Manufacturer and Model | $C_n$ (Ah)[1] | $C_m$ (Ah) | $B$ (A) | $n$ | $\delta^2$(%) |
|---|---|---|---|---|---|
| | | Equation (7) | | | |
| TrustFire IMR16340 | 0.65 | 0.610 | 5.385 | 3.242 | 4.000 |
| Aucooma IMR18650 | 2.8 | 2.807 | 25.912 | 3.326 | 2.627 |
| Keeppower IMR26650 | 4.2 | 4.033 | 27.169 | 3.161 | 2.753 |
| Samsung INR18650-13Q | 1.3 | 1.31 | 18.741 | 4.305 | 3.312 |
| Sony US18650VT3 | 1.5 | 1.49 | 27.208 | 4.41 | 2.776 |
| | | Equation (8) | | | |
| TrustFire IMR16340 | 0.65 | 0.614 | 6.644 | 5.028 | 2.520 |
| Aucooma IMR18650 | 2.8 | 2.807 | 31.866 | 5.513 | 1.837 |
| Keeppower IMR26650 | 4.2 | 4.037 | 33.684 | 5.262 | 1.741 |
| Samsung INR18650-13Q | 1.3 | 1.299 | 22.052 | 6.914 | 2.622 |
| Sony US18650VT3 | 1.5 | 1.494 | 31.799 | 7.062 | 1.61 |
| | | Equation (9) | | | |
| TrustFire IMR16340 | 0.65 | 0.620 | 6.748 | 3.041 | 0.510 |
| Aucooma IMR18650 | 2.8 | 2.833 | 32.222 | 13.543 | 1.076 |
| Keeppower IMR26650 | 4.2 | 4.073 | 34.027 | 14.864 | 0.542 |
| Samsung INR18650-13Q | 1.3 | 1.319 | 22.123 | 7.854 | 1.886 |
| Sony US18650VT3 | 1.5 | 1.494 | 32.122 | 10.676 | 0.789 |

[1] Nominal cell capacity. [2] Relative error of experimental data approximation by Equations (7)–(9).

Thus, from the coincidence of the experimental functions $f(i) = C(i/C_m)/C_m$ for the cells TrustFire IMR16340 and Aucooma IMR18650 (Figure 2a), it follows that these cells have the same electrodes and electrolytes, while the difference in the cell capacities is connected only with the different areas of their electrodes.

However, if the electrodes are constructively different (for example, having different thickness of active mass), the cells have electrodes of different types (for example, the cathodes IMR and INR) or different additions to their electrolytes and their active mass of electrodes were used [43,44], the experimental function $f(i) = C(i/C_m)/C_m$ for these cells must be different. This difference is seen for all other cells in Figure 2.

In Table 2, it is seen that Equations (7)–(9) correspond well to the experimental data at any discharge currents with the relative error of approximation—as a rule—less than 4%, which is sufficient for any practical purpose. Therefore, these equations correspond to the nature of the electrochemical processes at discharge of the lithium-ion cells. Thus, they can be used in various models for the cells SoC evaluation. However, it should be noted that, in all our experiments, the relative error of approximation of the experimental data by Equations (7)–(9) becomes less and less in the sequence: Equation (7), Equation (8), Equation (9). Thus, Equation (9) best corresponds to the experimental data (Table 2).

### 4.3. Analysis of the Use of the Peukert and Liebenow Equations for Lithium-Ion Cells

Now, consider the possibility of using the Peukert Equation (1) and the Liebenow Equation (2) for the lithium-ion cells. Neither Equation (1) nor Equation (2) can describe the experimental curves shown in Figure 2 over the entire variation range of the discharge current. For Equation (1), as the discharge current decreases, the cell released capacity tends to infinity, which does not make physical sense. Besides, at medium currents, the experimental curves are convex (Figure 2), while both Equations (1) and (2) give only concave curves (at positive values of all constants). As shown in Figure 2, the experimental curves are concave only starting from inflection point of the function $C(i)$ and keep this form to infinity, as well as at small discharge currents. Hence, it is only possible to use Equations (1) and (2) in these intervals of the discharge currents.

Let us check the possibility of using Equations (1) and (2) in these particular intervals of the discharge current values (Figure 3).

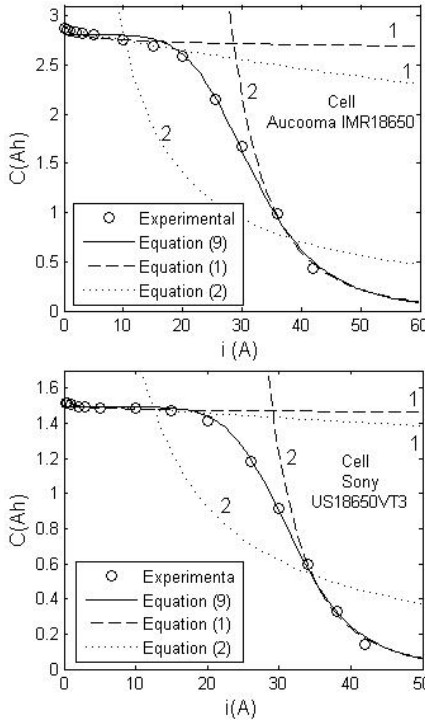

**Figure 3.** Comparison between experimental data for lithium-ion cells of different types and Equations (1), (2) and (9). Number 1 marks the Peukert and Liebenow equations at small discharge currents while Number 2 at high discharge currents.

　　The optimal parameters of Equations (1) and (2) for the lithium-ion cells in the interval of the discharge current values from the inflection point of the experimental curve to the highest currents used in experiments are presented in Table 3, and for small discharge currents in Table 4.

　　At small discharge currents, the function $C(i)$ is concave in the range from zero to the value of the discharge current at which function $C(i)$ begins to decrease sharply (Figure 3). This range is different for different cells. It depends on the type of electrodes used, the design of the cells, their capacity, etc. We conducted experimental studies in the range of discharge currents from $0.2C_n$ to the maximum currents $i_{max}$. The research results are presented in Table 4.

**Table 3.** Optimal parameters of Equations (1) and (2) for lithium-ion cells in the range of discharge currents from inflection point of experimental curve to the highest currents used in experiments.

| Manufacturer and Model | $C_n$ (Ah)[1] | $A$ | $n$ | $\delta^2$(%) |
|---|---|---|---|---|
| | Equation (1) | | | |
| TrustFire IMR16340 | 0.65 | 634.76 | 4.00 | 7.20 |
| Aucooma IMR18650 | 2.8 | $1.07 \times 10^7$ | 4.52 | 4.77 |
| Keeppower IMR26650 | 4.2 | $1.56 \times 10^7$ | 4.46 | 5.48 |
| Samsung INR18650-13Q | 1.3 | $8.88 \times 10^6$ | 5.28 | 6.72 |
| Sony US18650VT3 | 1.5 | $4.64 \times 10^8$ | 5.80 | 5.70 |
| | Equation (2) | | | |
| TrustFire IMR16340 | 0.65 | 92.42 | 47.81 | 39.4 |
| Aucooma IMR18650 | 2.8 | 791.32 | 28.10 | 30.8 |
| Keeppower IMR26650 | 4.2 | 887.50 | 21.36 | 22.8 |
| Samsung INR18650-13Q | 1.3 | 750.57 | 62.49 | 44.5 |
| Sony US18650VT3 | 1.5 | 984.76 | 53.06 | 39.6 |

[1] Nominal cell capacity. [2] Relative error of experimental data approximation by Equations (1) and (2).

**Table 4.** Optimal parameters of Equations (1) and (2) for lithium-ion cells at small currents, in the range of experimental discharge currents from 0.2Cn to imax.

| Manufacturer and Model | $C_n$ (Ah)[1] | $A$ | $n$ | $i_{max}$(A) | $\delta^2$(%) |
|---|---|---|---|---|---|
| | Equation (1) | | | | |
| TrustFire IMR16340 | 0.65 | 0.612 | 0.015 | 3 | 0.49 |
| Aucooma IMR18650 | 2.8 | 2.839 | 0.013 | 15 | 0.79 |
| Keeppower IMR26650 | 4.2 | 4.073 | $9.06 \times 10^{-3}$ | 15 | 0.67 |
| Samsung INR18650-13Q | 1.3 | 1.325 | 0.015 | 10 | 0.60 |
| Sony US18650VT3 | 1.5 | 1.504 | $6.73 \times 10^{-3}$ | 15 | 0.28 |
| | Equation (2) | | | | |
| TrustFire IMR16340 | 0.65 | 0.625 | 0.015 | 3 | 0.18 |
| Aucooma IMR18650 | 2.8 | 2.862 | $4.04 \times 10^{-3}$ | 15 | 0.33 |
| Keeppower IMR26650 | 4.2 | 4.099 | $2.94 \times 10^{-3}$ | 15 | 0.15 |
| Samsung INR18650-13Q | 1.3 | 1.334 | $4.58 \times 10^{-3}$ | 10 | 1.44 |
| Sony US18650VT3 | 1.5 | 1.509 | $1.78 \times 10^{-3}$ | 15 | 0.42 |

[1] Nominal cell capacity. [2] Relative error of experimental data approximation by Equations (1) and (2).

　　Equations (1) and (2) were obtained based on experimental investigations of the lead-acid batteries [30,31]. For lead-acid batteries, the Peukert equation is correct in the broad range of the discharge currents. Only at very small discharge currents, the Peukert equation does not correspond to the experimental data as the cell released capacity (in this equation) tends to infinity, which does not make physical sense. However, at all working discharge currents for lead-acid batteries, the Peukert equation corresponds to the experimental data quite well [30,31]. At low discharge currents for lead-acid batteries, the Liebenow equation corresponds well to experimental data [31]. Thus, the Peukert equation

and Liebenow equation complement each other when describing the released capacity of lead-acid batteries at any discharge currents.

For lithium-ion cells, the experimental dependence $C(i)$ of the cell released capacity on the discharge current differs significantly (Figure 2) from $C(i)$ for lead-acid batteries. For lithium-ion cells, the function C (i) is concave at small discharge currents, convex at medium discharge currents and again concave at high discharge currents. However, both Peukert Equation (1) and Liebenow Equation (2) describe only concave curves. Thus, theoretically, the Peukert Equation (1) and the Liebenow Equation (2) can describe the experimental function $C(i)$ (Figure 2) only from the discharge currents corresponding to the inflection point of the function $C(i)$ to infinity, as well as at small discharge currents. In these ranges of the discharge current values (according to our research results given in Figure 3 and Tables 3 and 4), indeed, the Peukert equation corresponds well to the experimental data. The relative error of approximation of the experimental data by Equation (1) is less than 7.2% (Table 3) for high discharge currents and less than 1% (Table 4) for small discharge currents.

The range of small discharge currents for some lithium-ion cells is quite large (Table 4), and it sometimes completely covers the usual working discharge currents of these cells. This is why many authors [18] use the Peukert equation to estimate the capacity of lithium-ion cells. Nevertheless, studies have shown that the scope of the Peukert equation for lithium-ion cells is limited.

The Liebenow Equation (2) poorly corresponds to the experimental data in the area of high discharge currents, where the function $C(i)$ is concave (Figure 3 (2) and Table 3). The relative error of approximation of the experimental data by Equation (2) is more than 22% (Table 3). However, in the range of small discharge currents, the Liebenow equation can be used to estimate the capacity of lithium-ion cells (Figure 3 (1) and Table 4).

*4.4. Improvement of the Hausman Analytical Model for Calculating the Remaining Capacity of Lithium-Ion Batteries Based on the Obtained Experimental Data*

In many studies [18–21,40], the cell remaining capacity is evaluated based on analytical models relying on the Peukert equation. We believe the most promising method is the analytical model of Hausmann [18]. This model is specifically designed to evaluate the remaining capacity of batteries in electric vehicles. In electric vehicles, batteries operate in dynamic mode, i.e., discharge currents can change quickly, and the temperature of the batteries must also be taken into account. A. Hausman's model [18] for calculating the remaining battery capacity has the form

$$C_t = C_m - \sum_{i=0}^{t} I_{eff}(i_i, T_i)\Delta t, I_{eff}(i_t, T_t) = f_1(i_t)f_2(T_t) = \gamma(i_t)^\alpha \left(\frac{T_{ref}}{T_t}\right)^\beta \tag{10}$$

where $C_m$ is an absolute capacity of a fully charged battery; $C_t$, $i_t$ and $T_t$ are the battery remaining capacity, current and temperature in an instant of time $t$, respectively; $T_{ref}$ is the reference temperature for an investigated battery; $\Delta t$ is the summing interval; and $\alpha$, $\beta$ and $\gamma$ are empirical constants.

In paper [18] the following values were used:

$$T_{ref} = 298K, \Delta t = 1s \tag{11}$$

In accordance with Equation (10), the equation

$$\Delta C_r(i) = f_1(i)f_2(T)\Delta t \tag{12}$$

shows that, as the discharge current increases, the removed effective capacity also increases. This correlation is in accordance with the commonly-used Peukert Equation (1) justification. In general, Equations (10) and (12) show that, when battery temperature decreases and load increases, the rate at which the remaining battery capacity is reduced also increases.

In addition, dividing the entire battery discharge time into the sum of small time intervals $\Delta t$ allows us to consider (under dynamic load) that at each time interval $\Delta t$ the current and temperature are constant. This makes it possible to use the empirical equations found for constant currents in Equation (10), in particular the Peukert equation, as shown below.

The model (10) was tested for commercial automotive-grade lithium batteries [18]. The maximal relative error received under diverse discharge dynamical modes did not exceed 5%.

The authors of [19] showed that the battery discharge capacity $C$ is related to the effective current $I_{eff}(i,T)$ by the equation

$$C = \frac{C_m}{I_{eff}(i,T)/i} \tag{13}$$

Thus, the analytical model of Hausmann [18] uses the Peukert equation of the form

$$C(i,\mathrm{T}) = \frac{A}{i^n}\left(\frac{T}{T_{ref}}\right)^{\beta}, n = \alpha - 1, A = C_m/\gamma \tag{14}$$

In Equation (14), the first multiplier is the classic Peukert Equation (1) and the second multiplier takes into account the dependence of the discharge capacity on temperature. The research presented in this manuscript showed that the classic Peukert Equation (1) can be used to calculate the discharge capacity of a lithium-ion battery in the range of currents from $0.2C_n$ to currents at which the battery discharge capacity begins to rapidly decrease (Figure 3(1)). This is a rather wide range of discharge currents, where the Peukert Equation (1) corresponds very well to the experimental data (Table 4). However, if the battery operates at high dynamic load in an electric vehicle, it is possible that the discharge currents will go beyond this range. In this case, the remaining battery capacity will not be calculated correctly. Therefore, in this case, instead of the classic Peukert Equation (1), it is necessary to use the generalized Peukert Equations (7)–(9), which are valid for any discharge currents (Figure 2 and Table 2).

From Equation (14), at a constant current $I = i0$, we obtain an equation for the dependence of the discharge capacity on the temperature in the form

$$C(T) = \mathrm{C_{mref}}\left(\frac{T}{T_{ref}}\right)^{\beta}, C_{mref} = A/i0^n \tag{15}$$

In Equation (15) at $T = T_{ref}$, the discharge capacity is $C = C_{mref}$ as in Equation (6). Of course, Equation (15) with an optimal choice of the empirical constants $C_{mref}$ and $\beta$ can describe the experimental dependence of the discharge capacity on temperature (Figure 1) over a fairly wide temperature range [18]. However, Equation (15) contains two fundamental disadvantages.

First, from dependency (15), it follows that the capacity $C$ reduces to zero at $T = 0$. However, from the electrochemical point of view, it is understandable that the capacity $C$ released by the battery reduces to zero much earlier. As the battery temperature decreases, the available capacity decreases due to increase of internal resistance of the battery and retardation of the chemical metabolism of the batteries effectively hindering the chemical reaction rate [13]. Thus, the capacity $C$ released by the battery is equal to zero at least at temperatures $T_L$ not less than an electrolyte congelation point.

Second, with increasing temperature, the discharge capacity $C$ of the battery in Equation (15) also increases unlimitedly. As the battery temperature increases, the internal resistance of the battery decreases and the chemical metabolism of the battery grows, thus effectively increasing the capacity of the battery [13]. Nevertheless, it is evident that the capacity cannot increase constantly with temperature growth. At reasonably large temperatures, active materials of electrodes start degrading, as does the electrolyte. Hence, there must exist a maximum temperature limit above which the capacity released by the battery ceases to increase.

Equation (15) does not take these two factors into account, while Equation (6) does. Therefore, from a theoretical point of view, Equation (6) more correctly describes the electrochemical processes

occurring in the battery when their temperature increases. Thus, Equation (6) can more accurately and over a wider range reflect the change in the discharge capacity of the batteries depending on the temperature. Thus, the conducted studies showed that, taking into account Equations (6)–(9), the analytical model of Hausmann [18] can be significantly improved. Indeed, if instead of the classic Peukert equation, a more correct (for lithium-ion batteries) generalized Peukert equation (for example, Equation (9)) is used, and, instead of Equation (15), the more correct Equation (6) is used, then, for the generalized Peukert equation, accounting for the temperature, we obtain from (14) the equation:

$$C(i, \text{T}) = \left( \frac{C_{mref}}{erfc(-B/n)} erfc\left( \frac{i-B}{n} \right) \right) K \frac{\left( \frac{T-T_L}{T_{ref}-T_L} \right)^{\beta}}{(K-1) + \left( \frac{T-T_L}{T_{ref}-T_L} \right)^{\beta}} \tag{16}$$

In Equation (16), the discharge capacity is $C = C_{mref}$ at $T = T_{ref}$ and $I = 0$.

Then, from Equation (16), taking into account Equation (13), for the effective current $I_{eff}$ $(i, T)$ (10) of the Hausmann [18] model, we obtain an improved equation.

$$I_{eff}(i, T) = i \frac{erfc(-B/n)}{erfc\left( \frac{i-B}{n} \right)} \left( 1 + \left( \frac{T_{ref} - T_L}{T - T_L} \right) (K-1) \right) \tag{17}$$

Thus, the use of Equations (7)–(9) and (6) (which most accurately reflect the electrochemical processes in lithium-ion batteries during their discharge) allows us to improve the analytical model by Hausmann [18]. This statement is also true for other analytical models of lithium-ion batteries containing the Peukert equation [20,21,40].

In conclusion, we note that, from the theoretical point of view, the most interesting is Equation (9). Equation (9) possesses its statistical basis (5) (complementary error function and therefore its parameters have statistical meaning) unlike Equations (7) and (8), which are just empiric equations.

The lithium-ion cells discharge process is a phase transition: it is the transition from phases of active mass of electrodes corresponding to cell's charged state to phases of the active mass of electrodes corresponding to the cell's discharged state. For example, in this study, for IMR lithium-ion cells during the charge–discharge cycle, the following phase transitions were observed regarding the active mass on the cathode and anode:

$$2MnO_2 + Li^+ + e^- \leftrightarrow LiMn_2O_4 (cathode) \tag{18}$$

$$LiC_6 \leftrightarrow C_6 + Li^+ + e^- (anode) \tag{19}$$

In physics [38], phase transitions are often described by the complementary error function (5), which is based on the normal law of distribution. On the level of molecules and ions, the discharge is a statistical process. Indeed, a statistical exchange process (for ions and molecules) is established at the interface between the active substance of the electrodes and the electrolyte in accordance with Equations (18) and (19), which is characterized by the exchange current $i_0$. While the discharge process at the interface of the active substance of the electrode and the electrolyte is described by the Butler–Volmer statistical function. Therefore, in general, on the level of molecules and ions, the charge–discharge process of cells is a statistical process. Thus, it is not surprising that the generalized Peukert equation is well described by the statistical function (complementary error function (5) or (9)). Hence, judging by the good coincidence of the experimental data with Equation (9) (Figure 2), it could be concluded that the cell's discharge is a statistical process subject to the normal law of distribution. This experimental fact, it seems to us, is significant for a theoretical explanation of the charge–discharge process in batteries as well as a better understanding of the Peukert equation.

## 5. Conclusions

Several conclusions can be drawn from the studies carried out.

First, for the lithium-ion cells, it is impossible to use the classic Peukert equation in the entire range of discharge currents as it does not correspond to experimental data for medium and very small discharge currents (Figure 3).

Second, the generalized Peukert Equations (7) and (9) correspond well to the experimental data at any discharge currents.

Third, among Equations (7)–(9), the most preferable ones for practical use are the generalized Peukert Equations (8) and (9). These equations approximate the experimental data with the least relative error (Table 2).

Fourth, research has shown that, when calculating the remaining capacity of a battery, its temperature must be taken into account. The dependence of the discharge capacity on temperature is proposed in [6], which correctly describes this dependence at all temperatures, both very low and high.

Fifth, on the basis of experimental studies, an improved analytical model by Hausmann is proposed for calculating the remaining capacity of lithium-ion batteries under dynamic loads. The proposed model is true for any discharge current in a wide temperature range.

Of course, the proposed analytical model requires further experimental and theoretical research, which is the topic of our further scientific studies.

**Author Contributions:** Conceptualization, N.E.G.; methodology, N.E.G., N.N.Y. and D.N.G.; software, N.E.G. and D.N.G.; validation, N.E.G., N.N.Y. and D.N.G.; formal analysis, N.E.G. and D.N.G.; investigation, N.E.G., N.N.Y. and D.N.G.; resources, N.E.G., N.N.Y. and D.N.G.; data curation, N.E.G. and N.N.Y.; writing—original draft preparation, N.E.G. and N.N.Y.; writing—review and editing, N.E.G. and D.N.G.; visualization, N.E.G. and N.N.Y.; supervision, N.E.G.; project administration, N.E.G.; and funding acquisition, N.E.G., N.N.Y. and D.N.G. All authors have read and agreed to the published version of the manuscript.

**Funding:** This research received no external funding.

**Conflicts of Interest:** The authors declare no conflict of interest. The funders had no role in the design of the study; in the collection, analyses, or interpretation of data; in the writing of the manuscript, or in the decision to publish the results.

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
