# Peer review of "A Critical Study of Using the Peukert Equation and Its Generalizations for Determining the Remaining Capacity of Lithium-Ion Batteries"

_applsci, doi:10.3390/app10165518_

Round 1

Reviewer 1 Report

The work by Galushkin et al. presents an interesting study for optimizing the obtaining of energy from lithium ion batteries. The topic is currently a hot topic for technology due to its many social implications. The work is well-driven and the results are sound. Therefore, I recommend the manuscript for publication after addressing some points:

-I recomment to the authors include the discussion within the presentation of the results to facilitate the understanding of the work.

-Experimental sectin should be more specific.

-No clear explanation is provided related to the absence of any impact of the temperature on the cell performance.

Reviewer 2 Report

The article addresses the current industrial research challenges on electric vehicle Li-ion batteries. It represents an interesting investigation on determining the remaining capacity of lithium-ion batteries. The manuscript can be improved by considering the following: 

  • Please check thoroughly sentence construction and mistakes. Few examples:
    • Line 42: …..open circuit voltage [6].
    • Line 44: Please rephrase the sentence staring ‘Besides, it is unsuitable …’
    • Line 50: ….. the SoC evaluation relies on …..
    • Line 223: “Now consider give consideration to the use possibility….”
    • Line 324: “For example, the generalized Peukert equations (9).”
  • A detailed literature review is needed to bring out the need for the present work. At present, authors have mentioned a few studies related to Peukert equation or other empirical models. Authors need to justify the advantages and challenges of these methods mentioned here. A comprehensive review is needed to identify the gaps in the literature and novelty of this work.
  • Please clarify the other battery types mentioned in line 87.
  • A flow chart can be produced for a better understanding of the experimental steps. And a figure for test set-up.
  • Please rephrase line 182-183.
  • Please elaborate why Eq. 9 was performed better than Eqs. 7 & 8.
  • Authors describe the research conducted by A. Hausmann. They need to provide evidence about the performance of the Eq. 17 to demonstrate that it’s an improved method.
  • Please restructure the conclusions point-wise identifying the key points. Also, add the future work and their implementation for real-time estimation.

Reviewer 3 Report

This is a well presented study and analysis of battery discharge modeling. It is clearly introduced and presented with insight into statistical interface charge transfer processes. This makes it more interesting beyond pure engineering discussion.

Would it be possible to extend discussion in prediction of the aging behaviour of batteries. the beta-exponents are related to the streatched exponential transport properties and their change is expected to be linked to the aging and change of interfaces. 

minor issues:

Cn is not defined in abstract

eqn 3 has typo in tanh
